# Predicting Kidney Transplantation Outcomes from Donor and Recipient Characteristics at Time Zero: Development of a Mobile Application for Nephrologists

**DOI:** 10.3390/jcm13051270

**Published:** 2024-02-23

**Authors:** Miguel Ángel Pérez Valdivia, Jorge Calvillo Arbizu, Daniel Portero Barreña, Pablo Castro de la Nuez, Verónica López Jiménez, Alberto Rodríguez Benot, Auxiliadora Mazuecos Blanca, Mª Carmen de Gracia Guindo, Gabriel Bernal Blanco, Miguel Ángel Gentil Govantes, Rafael Bedoya Pérez, José Luis Rocha Castilla

**Affiliations:** 1Nephrology Service, Hospital Virgen del Rocío, 41013 Sevilla, Spain; gabriel.bernal.sspa@juntadeandalucia.es (G.B.B.); miguelangelgentilgovantes@gmail.com (M.Á.G.G.); josel.rocha.sspa@juntadeandalucia.es (J.L.R.C.); 2Biomedical Engineering Group, University of Sevilla, 41092 Sevilla, Spain; jcalvillo@us.es; 3Department of Telematics Engineering, University of Sevilla, 41092 Sevilla, Spain; danporbar@alum.us.es; 4Regional Transplant Coordination of Andalusia, 41012 Sevilla, Spain; pablo.castro.sspa@juntadeandalucia.es; 5Nephrology Service, Hospital Regional de Málaga, 29010 Málaga, Spain; veronica.lopez.sspa@juntadeandalucia.es; 6Nephrology Service, Hospital Reina Sofía, 14004 Córdoba, Spain; alberto.rodriguez.benot.sspa@juntadeandalucia.es; 7Nephrology Service, Hospital Puerta del Mar, 11009 Cádiz, Spain; mauxiliadora.mazuecos.sspa@juntadeandalucia.es; 8Nephrology Service, Hospital Virgen de las Nieves, 18014 Granada, Spain; maria.gracia.sspa@juntadeandalucia.es; 9Pediatric Nephrology Service, Hospital Virgen del Rocío, 41013 Sevilla, Spain; rafael.bedoya.sspa@juntadeandalucia.es

**Keywords:** kidney transplantation, predictive model, graft failure, survival prediction, mobile app

## Abstract

(1) Background: We report on the development of a predictive tool that can estimate kidney transplant survival at time zero. (2) Methods: This was an observational, retrospective study including 5078 transplants. Death-censored graft and patient survivals were calculated. (3) Results: Graft loss was associated with donor age (hazard ratio [HR], 1.021, 95% confidence interval [CI] 1.018–1.024, *p* < 0.001), uncontrolled donation after circulatory death (DCD) (HR 1.576, 95% CI 1.241–2.047, *p* < 0.001) and controlled DCD (HR 1.567, 95% CI 1.372–1.812, *p* < 0.001), panel reactive antibody percentage (HR 1.009, 95% CI 1.007–1.011, *p* < 0.001), and previous transplants (HR 1.494, 95% CI 1.367–1.634, *p* < 0.001). Patient survival was associated with recipient age (> 60 years, HR 5.507, 95% CI 4.524–6.704, *p* < 0.001 vs. < 40 years), donor age (HR 1.019, 95% CI 1.016–1.023, *p* < 0.001), dialysis vintage (HR 1.0000263, 95% CI 1.000225–1.000301, *p* < 0.01), and male sex (HR 1.229, 95% CI 1.135–1.332, *p* < 0.001). The C-statistics for graft and patient survival were 0.666 (95% CI: 0.646, 0.686) and 0.726 (95% CI: 0.710–0.742), respectively. (4) Conclusions: We developed a mobile app to estimate survival at time zero, which can guide decisions for organ allocation.

## 1. Introduction

Renal transplantation is the treatment of choice for end-stage kidney disease [1,2]. Among its benefits are an evident improvement in quality of life, a removal of the requirement for dialysis, an increase in life expectancy, and cost-effectiveness [3,4]. Since the introduction of kidney transplantation in the 1960s, survival has notably improved, especially since the generalized use of cyclosporine, which has significantly decreased the graft rejection rate [5]. However, in recent years, different studies from countries belonging to the Eurotransplant Senior Program have questioned whether transplants always offer better survival [6,7]. These nonoptimal transplant outcomes are limited to older donors with controlled circulatory death. On the other hand, some studies claim that even with expanded or marginal donors, transplantation, including in elderly candidates, provides survival advantages over dialysis [8,9,10,11,12].

Obviously, the donor age has increased in recent decades, with progressively more comorbidities, which makes survival prognosis uncertain [13]. In fact, according to some studies, kidney graft survival has not improved in the last 20 years [14,15]. Therefore, personalized decision-making tools for kidney transplantation are of special importance. The tools available today do not differ significantly from those available 20 years ago and are mainly based on two aspects: the donor glomerular filtration rate and preimplant biopsy results. Preimplantation biopsy is a frequent cause of discarded organs that theoretically could have been implanted in certain recipients, offering acceptable outcomes, at least compared to dialysis. Currently, most studies state that there is no consistent association between biopsy findings and transplantation outcomes [16,17,18].

Predictive risk indices are another method used to assess donor and recipient features and have become relevant in recent years. The main predictive tool, attributed to its importance, is the Kidney Donor Profile Index (KDPI) implemented in the United States in 2014 within a new allocation policy [19,20,21,22]. This new index was created with the intention of assigning kidney grafts more efficiently to obtain the greatest longevity of the transplant and reduce retransplant rates. The particularity of this index with respect to previous ones is that it offers a quantitative scale of the risk of graft failure from a given donor using all grafts obtained in the previous year as the comparator. The KDPI has been combined with another index, that is, the estimated post-transplant survival (EPTS), to predict survival using four recipient variables.

After the KDPI, other indices were developed to estimate the prognosis of transplantation. One of the most recent tools is iChoose Kidney, a mobile application developed to help patients make informed decisions between dialysis and transplantation, as users are provided with survival estimates of between 1 and 3 years for both treatment modalities [23,24]. In recent years, “prognostic calculators” have been introduced using both parametric methods and artificial intelligence techniques, with the latter becoming an expanding field [25,26,27,28,29]. However, few of these tools include donor and recipient variables at the time of transplantation for shared decision making. Herein, we describe the development of a tool that allows for the estimation of graft and patient survival rates using data collected on the day of donation. We also developed an app for smartphones that could help physicians and patients choose the appropriate treatment.

## 2. Materials and Methods

### 2.1. Study Population

This was a retrospective, observational, cohort study that collected data from the Information System of the Transplant Autonomous Coordination of Andalusia (SICATA). A cohort of 5078 kidney transplants performed in Andalusia from 1 January 2006 to 31 December 2019 was studied. All cadaveric donor transplants performed during this period in individuals older than 18 years were included. Patients who underwent combined organ transplants were excluded from the study. The median follow-up was 62 months. The study was approved by the Ethics Committee of the Virgen Macarena-Virgen del Rocío University Hospitals and was conducted in accordance with the Declaration of Helsinki [30]. This study adhered to the STROBE guidelines for reporting observational studies [31].

### 2.2. Variables

The donor, recipient, and transplantation process variables that had an impact on survival were included. The donor characteristics analyzed were age, sex, cause of death, asystole, body mass index (BMI), expanded criteria, hypertension, diabetes, and KDPI. The following characteristics were analyzed for the recipient subjects: age, sex, diabetes, body mass index (BMI), time on dialysis, previous renal replacement therapy, hepatitis C, number of kidney transplants, Charlson index, coronary artery disease, and EPTS. Finally, the following transplant characteristics were included: number of human leukocyte antigen (HLA) incompatibilities, combined donor–recipient sex, and time in cold ischemia.

The outcomes evaluated were the patient and graft survival rates. Patient death after transplantation was counted with a functioning graft or until 90 days after returning to dialysis. Deaths that occurred 90 days after returning to dialysis were not attributed to the transplantation. Death-censored graft survival was defined as return to dialysis or re-transplantation. Because death and graft failure are considered independent events, a non-competing risk analysis was performed.

### 2.3. Statistical Analysis

The patients were randomly divided into training (70%) and validation (30%) cohorts. Continuous variables are expressed as means and standard deviations (SD) or medians and interquartile ranges (IQR; 25–75 percentiles) based on their distribution. The distribution of each variable included in the study was analyzed using the Kolmogorov–Smirnov test. Categorical variables are expressed as frequencies and percentages. The characteristics between cohorts were compared using the chi-squared test for categorical variables and a Student’s t-test or Mann–Whitney U test for continuous variables based on the distribution of the data. Graft and patient survival curves were calculated via the Kaplan–Meier method. The threshold for significance was set at *p* < 0.05.

### 2.4. Model Development

The association of each variable with transplant and patient survival was analyzed using univariate Cox regression. The results are expressed as hazard ratios, 95% confidence intervals, and corresponding *p* values. The assumption of proportional hazards was tested graphically by applying a log-minus-log survival plot for each variable.

Subsequently, multivariate analysis was performed using the Cox regression analysis with the conditional stepwise method for variable selection. The joint effect of pairs of variables was also assessed. Once significant variables were identified, bootstrapping was used, which is a statistical procedure that resamples a single dataset to create many simulated samples. Thus, the consistency of the results across samples (i.e., whether the identification of significant variables is exclusively dependent on the original sample) could be assessed. In cases in which the bootstrapping technique yielded results consistent with the identified model, the next step was to evaluate the validity of the model in the validation cohort. We also assessed the joint effects of the significant variables. Based on the results of the validated multivariate Cox regression analysis, two predictive models were developed: (1) death-censored graft survival and (2) patient survival.

Harrell’s C concordance index (c-statistic) was applied to evaluate the discriminatory power (the ability to separate patients with different prognoses). The calibration of the models was evaluated by comparing the observed cases (patient death or graft failure) with ones predicted by the corresponding model.

The statistical software packages used were SPSS 24.0 (IBM SPSS Statistics for Windows, version 24.0. IBM Corp., Armonk, NY, USA) and R version 4.1.0.

### 2.5. Mobile App

An Android mobile application and a service based on the REST interface were developed to facilitate the implementation of the models in clinical practice. Health professionals were involved in developing the first prototype. Appendix A shows screenshots of the Kidney Transplant App. The App code is available in [32].

## 3. Results

### 3.1. Demographic Data

The median age of the donors in the training cohort was 56 years (IQR 46–65 years, and the majority were men (61.3%) (Table 1). The main cause of donor death was stroke (56.6%), and brain death was the main donation method (82.7%). Forty-five percent of all donations were from expanded criteria donor (ECD). The median age of the recipients was 54 years (IQR 45–63 years). The treatment modalities prior to transplantation were hemodialysis (78.7%) and peritoneal dialysis (17.4%), with preemptive transplantation accounting for only 3.9%. Diabetes was a comorbidity in 13.3% of patients and a cause of primary kidney disease in 10% of patients. The median time on dialysis prior to transplantation was 593 days (IQR 205–980 days). The median cold ischemia time was 15.42 h (IQR 11.75–19.08 h). Regarding the combination of sexes between donors and recipients, the male donor/female recipient pairing was the most frequent, accounting for 39.2% of the cases (Table 1). The comparison of baseline characteristics between the two cohorts was not significant; thus, no bias was introduced with the split of the original population.

The evolution of populations regarding graft and patient survival is shown in Figure 1A,B, respectively.

### 3.2. Risk Prediction for Death-Censored Graft Survival

The donor age was significantly associated with death-censored graft survival (HR 1.021; 95% CI 1.017–1.024, *p* < 0.001). Uncontrolled donation after circulatory death was associated with a higher risk of graft loss (HR 1.576, 95% CI 1.213–2.047, *p* = 0.001), such as controlled donation after circulatory death of the donor (type III), which was associated with a lower graft survival (HR 1.567, 95% CI 1.355–1.812, *p* < 0.001). The degree of immunization (PRA%) was related to graft prognosis, increasing the risk of graft failure for each percentage increase in PRAs (HR 1.009, 95% CI 1.007–1.011, *p* < 0.001). Retransplant patients had a higher risk of graft loss (HR 1.494, 95% CI 1.335–1.648, *p* < 0.001) than those who underwent their first transplant.

The variables included in the final model for death-censored graft survival were donor age, donor circulatory death (controlled or uncontrolled), PRA, and re-transplantation (Table 2). After testing the proportional hazards assumption for each variable in the model, no violation was identified, and the variables did not show joint effects. The model was validated in an internal cohort that comprised 30% of the overall population. The model’s discriminatory power for the validation cohort was moderate (c-statistic, 0.666; 95% CI: 0.646, 0.686). To evaluate the calibration of the model, the cases were grouped according to the probability of graft failure. For each group, the failures estimated by the model were compared with those observed in the real population. Figure 2A shows the calibration plot of the graft survival model to our training population three years after transplantation. The calibration curves show a relevant similarity between observed and predicted cases, but the match is not perfect, which was confirmed by the value of the c-statistic.

### 3.3. Risk Prediction for Patient Survival

Age was the only donor characteristic significantly associated with patient prognosis (Table 3); for each additional year of donor age, the recipient mortality increased (HR 1.019, 95% CI 1.016–1.023, *p* < 0.001). Regarding recipient characteristics, age was significant only as a categorized variable. Thus, from the age of 40 years onwards, the risk of death also increased (age of the recipient 40–59 years: HR 1.848, 95% CI 1.52–2.23, *p* < 0.001; age older than 60 years: HR 5.507, 95% CI 4.524–6.704, *p* < 0.001). Similarly, diabetic nephropathy as a primary kidney disease (HR 2.089, 95% CI 1.887–2.311, *p* < 0.01) and time on dialysis before transplantation (HR 1.0003, 95% CI 1.0002–1.0003, *p* < 0.001) were significantly associated with death. Notably, being a male recipient (HR 1.229, 95% CI 1.135–1.332, *p* < 0.001) was a risk factor for mortality. Joint effects of the variables were not identified, and the proportional hazards assumption was satisfied. The c-statistic was 0.739 (95% CI: 0.729–0.749) for the training cohort and 0.726 (95% CI: 0.710–0.742) for the validation cohort, indicating that this model was generalizable to the regional survival data of kidney transplantation (Table 3).

To evaluate model calibration, the cases were grouped according to the probability of death. For each group, the correlation between deaths estimated by the model and those observed in the real population was established. Figure 2B shows the adequacy of the patient survival model for the training population. Similar to the graft survival model, it showed a good fit between observations and predictions. 

### 3.4. Translation of the Predictive Models to the Mobile Application

As an example (Appendix A), for a 38-year-old donor with uncontrolled circulatory death and a recipient with 0% PRAs who had never undergone a transplant, the graft survival in recipient 1 in the 1st, 5th, and 10th year would be 86%, 69%, and 50%, respectively. In contrast, if a patient with 90% PRAs is matched with the same donor for a fourth transplant (recipient 5), the graft survival would be 44%, 23%, and 12% at the 1st, 5th, and 10th year, respectively (Appendix A).

Donor age was the only variable that predicted patient survival. Among recipient characteristics, age, male sex, diabetic kidney disease, and time on dialysis were variables with prognostic importance. Appendix A shows the outcomes provided by the calculator for the graft from a 50-year-old donor implanted in a 50-year-old male recipient who had been on dialysis for 419 days (recipient 4). Patient survival in the first year would be 95%, 87% in the 5th year, and 70% in the 10th year. Using the same donor, but in a 60-year-old male recipient with diabetic kidney disease and 1025 days on dialysis (recipient 5), the survival rates would be 70%, 43%, and 23%, respectively (Appendix A).

## 4. Discussion

Mobile health technologies represent an opportunity to improve the information exchange between healthcare providers and patients. In this study, we developed a mobile tool with an adequate ability to predict graft and patient survival after kidney transplantation. It was based on a model composed of donor age, circulatory death (controlled and uncontrolled), PRAs, and re-transplantation, resulting in a c-statistic value of 0.666, which is practically identical to that of the KDPI for graft survival [21]. Regarding patient survival, the associated variables were the age of the donor and recipient, diabetic kidney disease, male sex, and dialysis duration. The c-statistic was 0.726, which is the same as that for the EPTS. Both the c-statistics and the fit of the models to our training population show that improvement is still possible. Future studies should explore and incorporate relevant variables into renal transplantation models to increase their predictive power.

Since the 2000s, different predictive models have been published to improve allocation policies for kidney transplantation [26,33,34,35]. Chronologically, the first index widely used in practice categorized donors as those with standard criteria (SCD) and those with expanded criteria (ECD) [36]. This subdivision between SCD and ECD has proven its usefulness in the so-called “old for old” allocation policies and still applies today. More recently, a key index owing to its subsequent influence has been the “Life Years From Transplant” (LYFT), by which, using donor and recipient variables, the estimated half-life with a functional graft compared to staying on dialysis was compared [37].

During the second decade of the 21st century, there has been new interest in the development of accurate tools for predicting the life of a graft or patient, driven by new allocation policies in the US. In 2014, a new organ allocation policy (Kidney Allocation System: KAS) was launched, based on two indices: the KDPI and EPTS. The KDPI is a continuous measure of organ quality. The ultimate objective of the KDPI is to classify the donor population by organ quality in an aggregate manner based on the risk conferred. The KDPI is a redefinition of the kidney donor risk index (KDRI) formula developed by Rao et al. [38], but with a lower number of variables. As a continuous measure, it approximates the quality of the organ better than the previous division between standard and expanded donors [10].

Comparing our index with the KDPI, the discriminatory powers were practically identical for both tools. Regarding the variables used, circulatory death was considered in both indices. In contrast, our multivariate analysis showed that donor hypertension and diabetes were not associated with graft survival. Using our tool, both re-transplantation and the degree of sensitization (PRAs) decreased (death-censored) graft survival. In contrast, EPTS and our model are very similar because they share most variables (age, time on dialysis, and diabetes) with the exception of re-transplantation, which is not significant in our model, and the prognostic association of male sex in our model.

One of the important findings of our work is the relevance of both uncontrolled and controlled circulatory death in death-censored graft survival, with the latter being particularly notable because it contradicts the most relevant published reports [39,40,41]. One explanation is that, during the study period, normothermic regional perfusion (NRP) was not generalized in the maintenance of these donors until 2019 [42]. Performing extraction techniques without NRP likely implies greater damage due to ischemia‒reperfusion and a worse prognosis. However, we did not have the donor warm ischemia time available for all cases during the study period; therefore, we were unable to adjust for this critical factor. Some studies have affirmed that warm ischemia for over 30 min is a risk factor for graft failure [43].

Another notable finding was the absence of an association between HLA mismatches and death-censored graft survival. HLA incompatibility has been considered a traditional risk factor, with a clearly favorable prognosis for very well-matched kidneys (0–2 HLA mismatches) [44,45]. However, the effect of mismatches is not linear (there are few differences between one and six HLA mismatches) and new evidence suggests that the influence of HLA incompatibility is less important. A recent study by the Australia–New Zealand registry (ANZDATA) with 7440 cadaveric donor kidney transplants performed in the period 2000–2018 concluded that a greater compatibility means greater benefits for graft and patient survival; however, the benefits were small and may be caused, in part, by a shorter stay on dialysis [46].

Our study highlights the impact of donor age on patient survival. Although we did not address waiting list mortality, the outcomes herein are consistent with the results reported by Hellemans et al. [7], who studied a cohort of transplants carried out in Belgium between 2000 and 2012, reporting that in elderly patients (>65 years), transplantation from an SCD donor does decrease mortality compared to dialysis, but any survival benefit with ECD transplantation versus dialysis may be small.

In our opinion, there is a lack of tools or decision aids that allow the adaptation of transplant prognosis information to specific candidates [47]. One of the existing calculators, iChoose [23], was developed by a multidisciplinary group that included physicians, epidemiologists, patients, and families with the aim of developing an easy-to-use instrument with “friendly” and easy-to-understand graphics. Subsequently, the same group studied the effect of iChoose on the knowledge of and access to transplantation in 470 patients. Although knowledge of the process increased in the intervention group (education + iChoose), access to transplantation was similar to that in the group in which only standard education was provided [24].

The main limitation of our study is its retrospective nature. Although the follow-up period was long (median follow-up, 62 months), a longer follow-up period may be needed for a better estimate of patient survival. Another limitation is that we were unable to examine whether initial immunosuppression, especially induction, had consequences on graft functionality and patient survival. For building the mobile device application, patient feedback was not included, although the usability of the app is currently being assessed. A further limitation is its restrained prediction level (C-statistics for graft survival of 0.666), which implies that its application should be made at the discretion of professionals; however, this prediction power is similar to that of the KDPI, which is a widely used index. Furthermore, its generalization to other populations, different to the one used here, should be monitored in further studies. Finally, real-time prediction of graft prognosis on the day of donation has limitations, fundamentally derived from the lack of many clinical variables that have an impact on the survival of the graft and the patient. The experience of the transplant team is essential for correctly interpreting the tool.

The strengths of this study include the large size and homogeneous sample, scrupulous follow-up of the recipients without any loss of vital status information, large number of variables, correction for confounding factors, use of resampling techniques, and internal validation of the results. Contrary to other risk indices, our study included patients who underwent re-transplantation.

In conclusion, the results of our study indicate that kidney transplantation prognosis can be predicted with an acceptable reliability. The development of a mobile tool (Kidney Transplant App) can help patients and professionals make joint decisions based on survival expectations provided by the tool.

## Figures and Tables

**Figure 1 jcm-13-01270-f001:**
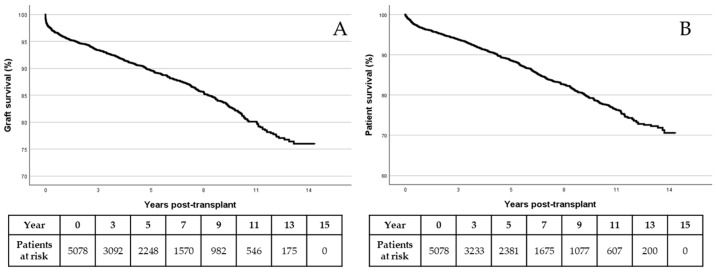
Graft (**A**) and patient (**B**) survival by Kaplan–Meier.

**Figure 2 jcm-13-01270-f002:**
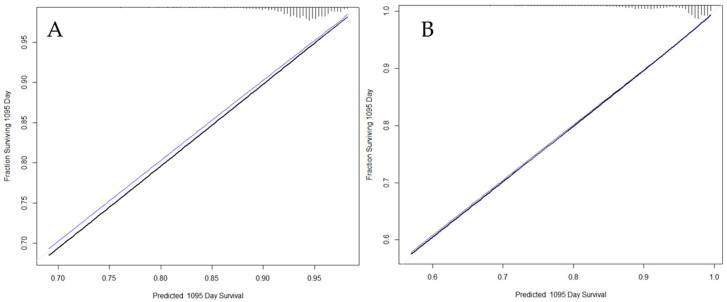
Calibration plots of the graft survival model (**A**) and patient survival model (**B**) to our training population after 3 years. Black line: resulting calibration plot; Blue line: bias-corrected calibration plot; Gray line: ideal calibration.

**Table 1 jcm-13-01270-t001:** Demographic and clinical characteristics.

Variables	Training Cohort (N = 3559)	Validation Cohort (N = 1519)	*p*
Donor characteristics			
Age (median, IQR)	56 (46.5–65.5)	57 (47.5–66.5)	0.25
Sex (*n*, %)			0.99
Male	2183 (61.3%)	932 (61.4%)	
Female	1376 (38.7%)	587 (38.6%)	
Cause of death (*n*, %)			0.26
Head Trauma	605 (17.0%)	238 (15.7%)	
Stroke	2014 (56.6%)	869 (57.2%)	
Other	940 (26.4%)	412 (27.1%)	
Donation after circulatory death (*n*, %)			0.25
No (DBD)	2943 (82.7%)	1233 (81.2%)	
Uncontrolled DCD	114 (3.2%)	57 (3.7%)	
Controlled DCD	502 (14.1%)	229 (15.1%)	
BMI (*n*, %)			0.44
Underweight (≤18.5 kg/m^2^)	39 (1.1%)	13 (0.9%)	
Normal (18.5 to <25 kg/m^2^)	929 (26.1%)	380 (25.0%)	
Overweight (25 to <30 kg/m^2^)	1437 (40.4%)	634 (41.7%)	
Obese (≥30 kg/m^2^)	697 (19.6%)	283 (18.6%)	
NA	457 (12.8%)	209 (13.8%)	
Donor (*n*, %)			0.46
SCD	1949 (54.8%)	794 (52.3%)	
ECD	1610 (45.2%)	725 (47.7%)	
Hypertension (*n*, %)	1037 (29.1%)	478 (31.5%)	0.1
Diabetes (*n*, %)	235 (6.6%)	102 (6.7%)	0.88
KDPI (median, IQR)	49.5 (24.5–74.5)	51.2 (26.5–75.9)	0.12
Recipient characteristics			
Age (median, IQR)	55 (45.7–64.2)	54 (45.0–63.0)	0.78
Sex (*n*, %)			0.87
Male	2255 (63.4%)	966 (63.6%)	
Female	1304 (36.6%)	553 (36.4%)	
Diabetes (*n*, %)	472 (13.3%)	230 (15.1%)	0.08
BMI (*n*, %)			0.79
Underweight (≤18.5 kg/m^2^)	47 (1.3%)	20 (1.3%)	
Normal (18.5 to <25 kg/m^2^)	935 (26.3%)	405 (26.7%)	
Overweight (25 to <30 kg/m^2^)	928 (26.1%)	397 (26.2%)	
Obese (≥30 kg/m^2^)	535 (15.0%)	227 (14.9%)	
NA	1114 (31.3%)	470 (30.9%)	
Primary kidney disease (*n*, %)			0.36
Diabetes	356 (10.0%)	165 (10.9%)	
Other	3203 (90.0%)	1354 (89.1%)	
Time on dialysis (median, IQR)	593 (205.5–980.5)	619 (203.2–1034.7)	0.44
Previous kidney replacement treatment (*n*, %)			0.71
Pre-emptive kidney transplant	138 (3.98%)	58 (3.8%)	
HD	2800 (78.7%)	1203 (79.2%)	
PD	621 (17.4%)	258 (17.0%)	
Hepatitis C (*n*, %)	105 (2.9%)	39 (2.6%)	0.48
EPTS (median, IQR)	49.8 (25.0–74.6)	50.4 (25.0–75.8)	0.31
Kidney transplant number (median, IQR)	1 (1–1)	1 (1–1)	0.77
Charlson index (*n*, %)			0.65
0–3	2203 (61.9%)	914 (60.2%)	
>4	1356 (38.1%)	605 (39.8%)	
Coronary disease (*n*, %)	157 (4.4%)	73 (4.8%)	0.54
Transplant characteristics			
Number of HLA, mm (median, IQR)	4 (3–5)	4 (3–5)	0.72
Sex combinations (*n*, %)			0.94
MD/MR	1395 (39.2%)	597 (39.3%)	
MD/FR	788 (22.1%)	335 (22.1%)	
FD/MR	860 (24.2%)	369 (24.3%)	
FD/FR	516 (14.5%)	218 (14.3%)	
Cold ischemia time (hours) (median, IQR)	15.42 (11.75–19.08)	15.50 (11.83–19.07)	0.56

BMI, body mass index; DBD, donation after brainstem death; DCD, donation after circulatory death; ECD, expanded criteria donor; EPTS, estimated post-transplant survival; FD, female donor; FR, female recipient; HD, hemodialysis; HLA, human leukocyte antigen; IQR, interquartile range; KDPI, kidney donor profile index; MD, male donor; mm; mismatches; MR, male recipient; NA, not available; PD, peritoneal dialysis; SCD, standard criteria donor; SD, standard deviation.

**Table 2 jcm-13-01270-t002:** Factors associated with death-censored graft survival.

	Multivariate Analysis			95% CI Sample	
Variables	Hazard Ratio (95% CI)	*p*	Β-Coefficient	Lower	Upper
Donor characteristics					
Age (continuous)	1.021 (1.017–1.024)	<0.001	0.021	0.017	0.024
Asystolic donor					
No (DBD)	1	-	-	-	-
Uncontrolled DCD	1.576 (1.213–2.047)	0.001	0.455	0.185	0.704
Controlled DCD	1.567 (1.355–1.812)	<0.001	0.449	0.301	0.591
Recipient characteristics					
Panel reactive antibodies	1.009 (1.007–1.011)	<0.001	0.009	0.007	0.011
Number of kidney transplants	1.494 (1.355–1.648)	<0.001	0.402	0.288	0.507

c-statistic discrimination test = 0.666. CI, confidence interval; DBD, donation after brainstem death; DCD, donation after circulatory death.

**Table 3 jcm-13-01270-t003:** Factors associated with patient survival.

	Multivariate Analysis			95% CI Sample	
Variables	Hazard Ratio (95% CI)	*p*	Β-Coefficient	Lower	Upper
Donor characteristics					
Age (continuous)	1.019 (1.016–1.023)	<0.001	0.019	0.016	0.023
Recipient characteristics					
Age					
<40 (REF)	1	-	-	-	-
40–59	1.848 (1.529–2.234)	<0.001	0.620	0.429	0.820
60+	5.507 (4.524–6.704)	<0.001	1.716	1.521	1.911
Sex					
Female (REF)	1	-	-	-	-
Male	1.229 (1.135–1.332)	<0.001	0.226	0.151	0.315
Primary kidney disease					
Other (REF)	1	-	-	-	-
Diabetes	2.089 (1.887–2.311)	<0.001	0.802	0.711	0.896
Time on dialysis (continuous)	1.000263 (1.000225–1.000301)	<0.001	0.000263	0.000223	0.000304

c-statistic discrimination test = 0.726. CI, confidence interval; REF, reference.

## Data Availability

The data underlying this article will be shared upon reasonable request to the corresponding author.

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
