# Peer review of "Predicting Kidney Transplantation Outcomes from Donor and Recipient Characteristics at Time Zero: Development of a Mobile Application for Nephrologists"

_jcm, 2024, doi:10.3390/jcm13051270_

Round 1

Reviewer 1 Report

Comments and Suggestions for Authors

This is a retrospective study identifying factors associated with recipient and graft survival. This is an attempt at predicting patient and graft survival using data at the start with results similar to KDPI. My query is whether this could be reproducible in cohorts with higher representation of head trauma  as a cause of death and lesser number of expanded criteria donors and whether this can replace the KDPI. If not what would be the novelty in comparison to the KDPI. Again decision for going in for a transplant would be based on comparison with remaining on dialysis and this may be worth mentioning

Reviewer 2 Report

Comments and Suggestions for Authors

interesting perspective and clearly add tools for rejection prediction. implementation to other centres should be performed carefully, and laregr study is required before the tool can be used in organ allocation protocol

Reviewer 3 Report

Comments and Suggestions for Authors

Congratulations to all authors to this great work.

It ist presented clearly, structured and well designed.

Mentiones nearly all relevant pro´s and con´s of methodology and results.

The parameters on both sides (donor and receipient) are chosen on a daily relevant basis.

A small addition:

There can be many more parameters that could be included into the calculation. So one limitation ist the quite huge discrepancy of estimated and observes cases (page 7) in the middle field (Fig 1).

Here you should add one or two more sentences, to highlight that the decision making for an organ acception is sometimes not clearly predictable (missing prospective data as mentioned) but is still an approach with the model to date that should always be evaluated by an experienced doctor.

Reviewer 4 Report

Comments and Suggestions for Authors

In this paper the authors aim at producing two predicting models that allows for the estimation of patient and graft survival at the moment of kidney transplantation with the additional bonus of providing a mobile interface, that can be used by medical doctors and patients alike. The prediction capabilities are in the range of previous models. 

As a disclosure I am a statistician that has worked in the field of kidney transplantation, I do not have a deep knowledge on the clinical variables themselves and can only hope that another reviewer can provide for a more critical eye on this respect. I can say that the variables considered, and retained, are in line with what I have encountered in the literature. 

My comments 

1.     It is a predictive model, I expected a mention of prediction metrics (C-index) in the abstract. Also, in the abstract there’s no mention that this is related to kidney transplant.

2.     The introduction gives a general overview, but there are currently other prediction models available, such as Loupy et al (2019) https://doi.org/10.1136/bmj.l4923 , even though their model encompasses post-transplantation variables.

3.     Better description of the dataset(s), such a Kaplan-Meier with numbers at risk so that we have a clear view of the number of patients who are still included in the study, those who have not yet experienced the event or been lost to follow-up, at each time interval. The follow up time is mentioned only in the discussion.

4.     When mentioning the statistical analysis, comparing “the two cohorts” by means of univariate tests on the variables gives the impression that we have two real cohorts, whilst this is a resampling artefact that there are 2 cohorts. Personally, I would have had performed the training/test split step N times and pooled the results.

5.     The final set of predictors is not a surprising group, as mentioned by the authors, all are known to be associated with the outcomes. Nevertheless, I do have some issues with the variable selection process. It includes bootstrapping, producing some sort of replication, but it still is a sort of stepwise process if I understood correctly? I might be assuming things because it is not completely clear how it was done. It was: univariate Cox-> selection of the significant ones-> examine effect (interaction?) of pairs of variables (“joint effect of the pairs of variables was also assessed”)-> selection of significant ones-> all previously selected fitted simultaneously in the same final model (Tables 2 and 3).  There are currently different methodologies for variable selection, such as penalized regression that considers all variables at the same time (can include bootstrapping as well). 

6.     Figure 1 is an attempt of a calibration plot, I am not familiar with this particular rendition. Function calibrate() from the packge “rms’ in R is perfect for this.

7.     I find this paragraph confusing (line 102):

“The outcomes evaluated were the patient and graft survival rates. Patient death after transplantation was counted with a functioning graft or until 90 days after returning to dialysis. Deaths that occurred 90 days after returning to dialysis were not attributed to the transplantation. Death-censored graft survival was defined as return to dialysis or retrans-plantation. Because death and graft failure are considered independent events, a non-competing risk analysis was performed.”

 Outcomes: “Patient survival” and ‘Graft survival”

“Patient death after transplantation was counted with a functioning graft or until 90 days after returning to dialysis.”  

My interpretation is that if a patient died with a functioning graft it was counted as an event of the outcome “patient survival’

“Deaths that occurred 90 days after returning to dialysis were not attributed to the transplantation.” 

 Meaning that deaths after this period where not an event for the ‘patient survival’ outcome  (the event for “graft survival’ outcome has already occurred because they are back to dialysis)

 “Death-censored graft survival was defined as return to dialysis or retransplantation.”

I interpret this as:  Graft failure was defined as loss of graft function, return to dialysis or retransplantation. In case of death with a functioning graft, we censored graft survival at time of death. If this is the case, this is a competing risks scenario. I was pleased to see that the authors were aware of this issue, however I am not able to understand the argument for it not being a competing risks case for the “graft failure” outcome. In survival analysis, a competing risk is an event whose occurrence precludes the occurrence of the primary event of interest, so if a patient dies with a functional graft it precludes the possibility of an event (graft failure).

Comments on the Quality of English Language
